# Surface Cleaning Effect of Bare Aluminum Micro-Sized Powder by Low Oxygen Induction Thermal Plasma

**DOI:** 10.3390/ma15041553

**Published:** 2022-02-18

**Authors:** Dasom Kim, Yusuke Hirayama, Kenta Takagi, Hansang Kwon

**Affiliations:** 1Department of Materials System Engineering, Pukyong National University, 365, Sinseon-ro, Nam-gu, Busan 48513, Korea; ds-kim@aist.go.jp; 2Magnetic Powder Metallurgy Research Center, The National Institute of Advanced Industrial Science and Technology (AIST), 2266-98, Anagahora, Shimoshidami, Moriyama-ku, Nagoya 463-8560, Japan; hirayama.yusuke@aist.go.jp; 3Department of Research and Development (R&D), Next Generation Material Co., Ltd., 365, Sinseon-ro, Nam-gu, Busan 48547, Korea

**Keywords:** low oxygen induction thermal plasma, surface cleaning, electrical conductivity

## Abstract

The development of bare metal powder is desirable for obtaining conductive interfaces by low-temperature sintering to be applied in various industries of 3D printing, conductive ink or paste. In our previous study, bulk Al made from Al nanopowder that was prepared with low-oxygen thermal plasma (LO-ITP), which is the original metal powder production technique, showed high electrical conductivity comparable to Al casting material. This study discusses the surface cleaning effect of Al particles expected to be obtained by peeling the surface of Al particles using the LO-ITP method. Bare metal micro-sized powders were prepared using LO-ITP by controlling the power supply rate and preferentially vaporizing the oxidized surface of the Al powder. Electrical conductivity was evaluated to confirm if there was an oxide layer at the Al/Al interface. The Al compact at room temperature produced from LO-ITP-processed Al powder showed an electrical conductivity of 2.9 · 10^7^ S/m, which is comparable to that of cast Al bulk. According to the microstructure observation, especially for the interfaces between bare Al powder, direct contact was achieved at 450 °C sintering. This process temperature is lower than the conventional sintering temperature (550 °C) of commercial Al powder without any surface cleaning. Therefore, surface cleaning using LO-ITP is the key to opening a new gate to the powder metallurgy process.

## 1. Introduction

Recently, interest in the low-temperature bonding of metal powders has grown in applications such as 3D printing to manufacture exquisite electronics [1,2], conductive ink [3,4,5], conductive paste for bonding [6,7], magnetic data storage devices [8] and metal matrix composites [9,10]. There are several advantages of low-temperature bonding of metal powders: (1) reduced energy consumption, (2) the suppression of interfacial reactants and (3) the prevention of damage to thermal-sensitive bonding counterparts. Solid-state bonding (sintering) could be performed at a temperature that is over 80% of the temperature typically used in conventional sintering [11,12]. To decrease the sintering temperature, direct contact between metal powders is required. This is because the oxide film, which is always formed on the surface of metal powders, has a much lower thermal diffusivity than its metal inclusions [13]. In particular, the diffusivity of easily oxidizable metals, such as Al and Mg, is greatly degraded by the oxide layer between metal powders compared to other metals, such as Cu and Fe [14]. For example, it has been reported that 550 °C or higher is required as a sintering temperature for Al [15,16,17,18] because an amorphous oxide film formed natively on the surface at room temperature can be broken after crystallization by heating over 550 °C [19]. The surface oxide film prevents direct contact between metal particles, resulting in (1) an increase in the bonding temperature and (2) the deterioration of intrinsic properties of metal powder, such as high electrical conductivity.

Although oxide-free Cu [20] and Ag [21] particles can be prepared through the hydrogen reduction process, most oxidizable metals, such as Al and Mg, cannot be prepared by reducing the surface oxide layer using hydrogen. Although the partial oxide-free Al/Al interface was observed after spark plasma sintering of Al powder, the oxide-free interface could be formed at high temperature of 540 °C [18]. When observing the Al/Al interface in Al compacts prepared by spark plasma sintering at 400 °C using oxidized Al powder, the uniform oxide layer was formed [22], indicating that the cleaning effect of spark plasma can be ignored at low temperatures. Our research group developed a new bottom-up fine bare metal powder production technique called “low oxygen induction thermal plasma (LO-ITP)” [23] which can prepare metal and metal alloy powders [24,25,26]. The thickness of the surface oxidation layer of the nanopowder prepared by the LO-ITP process is thin enough not to lose conduction. In particular, the Al and Al particles of an Al sample with a nanostructure bulked at 400 °C achieved direct bonding. As a result, the relative density was 95%, achieving 2.9 × 10^7^ S/m, which is the same level of electrical conductivity (EC) as the Al cast material [23].

The nanoparticles obtained by LO-ITP pass through the gas phase, and then undergo uniform nucleation and heterogeneous coagulation to form nanoparticles. When the process is carried out in an inert gas atmosphere, in principle, the oxygen contained in the gas and the raw material powder contributes to the formation of the oxidized layer of the nanoparticles. Therefore, when nanosized powder is synthesized from micro-sized raw material powder, the surface area will be larger. For example, if the particle size is reduced to 1/100, the surface area will be 10,000-times larger, andhe thickness of the oxide layer of the obtained nanoparticles will be 10,000-times smaller. Therefore, nanoparticle synthesis with a very thin oxide layer can be realized.

Is there a way to enable similar direct contact with micro-sized particles? From the reported results, direct contact is possible when the oxide layer on the surface can be removed. As mentioned earlier, Al oxide cannot be reduced by hydrogen reduction. Therefore, other methods must be considered. According to R.H. Lamoreaux et al., the vapor pressure of Al-O exceeds the vapor pressure of Al at 2230 K, and the oxygen partial pressure is 10^−10^ atm [27]. Therefore, it is conceivable that Al-O evaporates preferentially when the temperature is raised to a high temperature under a low oxygen partial pressure. Since the oxygen partial pressure of Ar G1 gas used in the thermal plasma process [23] is 10^−7^ atm or less, it is possible that the vapor pressure of Al-O exceeds the vapor pressure of Al even during the actual process. Therefore, Al-O is preferentially vaporized by the high-temperature plasma, and only the Al core remains. In this paper, this phenomenon is referred to as the surface cleaning effect. In this study, it was verified whether the cleaning effect of the surface of the micro-sized particles could be obtained by performing high-temperature heat treatment under the condition that the raw material micro-sized Al particles were not completely evaporated using the LO-ITP process. Here, only the surface cleaning effect of thermal plasma was considered. Other factors that can remove the oxide layer, such as the spark plasma cleaning effect, were ignored, since the LO-ITP-processed Al powder was consolidated by current sintering at a low temperature range (~400 °C). To confirm if the surface cleaning effect occurred, the oxygen content and electrical conductivity of Al compacts consolidated by cold compaction and sintering at low temperature from Al particles were evaluated. Finally, the microstructure of the Al-Al particle interfaces was observed.

## 2. Materials and Methods

Figure 1 shows a schematic illustration of the experimental procedure. Commercial Al powder (particle size 20 μm, purity 99.99%, oxygen level 0.14 wt.%, Kojundo Chemical Lab. Co., Ltd., Saitama, Japan) was used as a raw material. The plasma was prepared by setting the radio frequency (RF) powder to 3 kW in a chamber filled with Ar gas (G1 grade, oxygen level less than 0.1 ppm) with a pressure of 70 kPa. Then, raw material was inserted into the chamber from the top of the torch by a powder feeding system (TP-99010FDR, JEOL Co., Ltd. Tokyo, Japan) at a feed rate of up to 5 g/min through the carrier gas. The Al powder processed with LO-ITP was collected from the main chambers and the Cu plates in a glove box where the oxygen content was controlled to be under 0.5 ppm. By measuring the specific surface area using the Brunauer–Emmett–Teller measurement (BET, 3 Flex Physisorption, Micrometrics Instrument Corp. Norcross, GA, USA) with nitrogen gas, the average particle size of the LO-ITP processed Al powder was calculated. The particle size distribution of the processed Al powder was estimated by field emission scanning electron microscopy (FE-SEM) (TESCAN, VEGA II LSU, Brno, Czech Republic). As a reference, we gradually oxidized the Al powder processed with LO-ITP by exposing it to 1% O_2_-Ar gas for 12 h (denoted as “exposed Al powder”). Conversely, the Al powder processed with LO-ITP that had not undergone gradual oxidization was called the “unexposed Al powder.”

In the glove box where oxygen content was controlled under 0.5 ppm, the unexposed Al powder was poured into a tungsten cobalt mold, and the mold set was transferred to the current sintering machine using a transfer vessel filled with heptane to avoid further oxidation. The consolidation of exposed Al powder was prepared under the same procedures in air and without protection from further oxidization, such as the usage of heptane. Consolidation was conducted by a current sintering machine in a vacuum atmosphere without heating (no application of current), that is, cold compaction or with heating (applying current) under a compaction pressure of 300 MPa. When conducting sintering, the inside temperature of the mold near the Al powder was measured by a thermocouple. The samples were heated to an elevated process temperature of 200 °C or 450 °C with a heating rate of 40 °C/min and held for 1 min. The sintered mold was cooled natively in a vacuum atmosphere. In this paper, the Al compacts prepared by unexposed and exposed Al powder were called “unexposed Al compact” and “exposed Al compact”, respectively. The density of the Al compacts was measured using the Archimedes method with a densitometer (KERN ABJ, 120–4 M, Frankfurt, Germany). The relative density was calculated from the full density by considering the volume fractions of Al and Al oxide. The volume fraction of Al oxide (assuming Al oxide as Al_2_O_3_) was calculated by an equation [22] using the oxygen level of the consolidated Al compacts measured by an Oxygen analyzer (EMGA-620 W, HORIBA, Ltd., Kyoto, Japan):(1)$$V_{Al_{2}O_{3}}=W_{O}\cdot M_{Al_{2}O_{3}}/\left\{W_{O}\cdot M_{Al_{2}O_{3}}+\rho_{Al_{2}O_{3}}\cdot \left(3M_{O}\cdot W_{Al}-2M_{Al}\cdot W_{O}\right)/\rho_{Al}\right\}$$
where $W_{O}$ and $W_{Al}$ are the weight percentages of O and Al, respectively. $\;M_{Al_{2}O_{3}}$, $M_{O}$ and $M_{Al}$ are the molar masses of *Al_2_O_3_*, O and Al, respectively; and $\rho_{Al_{2}O_{3}}$ and $\rho_{Al}$ are the densities of Al_2_O_3_ and Al, respectively.

The oxide layer thickness ($t_{Al_{2}O_{3}}$) was estimated by an equation:(2)$$t_{Al_{2}O_{3}}=\left\{D\left(1-\sqrt[{3}]{1-V_{Al_{2}O_{3}}}\right)\right\}/2$$
where *D* is diameter of particles. The EC of consolidated Al compacts was evaluated using an eddy current conductivity tester (TMD-102, TM Teck, Beijing, China) and a 4-probe method using a Loresta-GX MCP-T700 (Mitsubishi Chemical Analytech Co., Ltd., Tokyo, Japan) at room temperature. The cross-section of unexposed Al compacts was observed by FE-SEM (JSM-7800F, JEOL, Tokyo, Japan).

## 3. Results and Discussion

Figure 2 shows FE-SEM images of the Al powder processed with LO-ITP. There are two types of particles that are divided by particle size: large particles and fine particles, as shown in Figure 2a,c, respectively, with spherical shapes. From the FE-SEM images, the mean particle size and the distribution of the large particle was estimated to be 16.8 µm and σ = 6.5 µm, respectively. The mean particle size and the distribution of the fine particle was estimated to be 0.2 µm and σ = 0.1 µm, respectively. These values were determined by fitting to the log-normal distribution function. Comparing the average particle size obtained by measuring the surface area (2.0 m^2^/g) from the BET, measurement with the average particle size obtained from the SEM, the number of fine particles obtained is estimated to be 17 wt.%.

In Figure 3, the correct relative density is plotted against the process temperatures of 20 °C, 200 °C and 450 °C for unexposed and exposed Al compacts. Here, since the density of Al oxide (3.9 g/cm^3^) is 1.4 times higher than that of Al, the relative density was corrected by calculating the amount of oxide from the measured oxygen content of Al compacts using Equation (1), as shown in Table 1. The correct relative density of unexposed compact Al was nearly saturated at a process temperature of 200 °C. On the other hand, the correct relative density of exposed compact Al was not saturated at 450 °C.

Since the EC of Al oxide is 10^−19^ times that of Al, the EC was evaluated to confirm whether the Al oxide layer exists at the Al/Al interfaces. Figure 4 shows the EC as a function of the process temperature for the unexposed and exposed Al compacts prepared in this study. For comparison, the EC results of the Al bulk body prepared using commercial Al powder (denoted as “raw Al compact”) are also shown [15,16]. Here, the effect of particle size on EC was not overestimated, assuming that the quoted EC values were not underestimated because the commercial Al powders used had a larger particle size than the LO-ITP-processed Al micro-sized powder. The EC of the raw Al compact was enhanced with increasing process temperature and reached the EC value of cast Al at process temperatures beyond 550 °C. However, the EC of 2.4∙10^7^ S/m for the unexposed Al compacts prepared at room temperature was comparable to that of cast Al bulk. In the case of exposed compact Al, the EC reached approximately half of that of unexposed Al compacts. The oxide layer thickness was calculated using Equation (2) by measuring the oxygen content. Assuming that the Al powder was composed of large Al particles with a particle size of 16.8 µm (83 wt.%) and fine particles with the particle size of 0.2 µm (17 wt.%), the oxide layer thicknesses in the unexposed and exposed Al compacts were 1.9 nm and 3.5 to 6.8 µm, respectively. Based on the above hypothesis that the oxide has preferential evaporation by heat treatment at an ultra-high temperature under a low oxygen partial pressure to obtain a cleaning effect on the surface of Al particles, a much higher content of oxide (oxygen) is contained in the obtained nanopowder. Therefore, it is possible the thickness of oxide layer on the surface of the micro-sized Al particles, which occupy most of the obtained powder, is sufficiently thin compared to that obtained by the arithmetic mean of 1.9 nm, which might lead the sufficient conduction [22,28,29].

The arbitrary interface between Al powder was observed by FE-SEM in an Al compact prepared by cold compaction (Figure 4b), sintering at 200 °C (Figure 4c) and sintering at 450 °C (Figure 4d). The area where the Al particles were directly contacted is indicated by arrows in the figures. As the process temperature increased, so did the frequency of the direct contacted area. As a result, the increase in electrical resistance due to the non-direct contacted interface could be suppressed. Therefore, the conductivity could be kept high. On the other hand, when using commercially available Al particles, it is necessary to remove the oxide layer on the surface to obtain high conductivity. This requires a temperature above 550 °C [18,19]. Therefore, the oxide film of ITP-processed Al powder is thin enough to secure conduction, that is, it has a surface cleaning effect. To evaluate the oxide film effect between the grains directly, high-resolution microstructure analysis, such as TEM (JEOL Ltd. Tokyo, Japan), 3D atom probe and in-situ TEM observation at an elevated temperature [30], are effective. These are the next tasks.

## 4. Conclusions

We succeeded in obtaining the surface cleaning effect of Al micro-sized powder by preferentially evaporating the nonconductive surface oxide layer through the LO-ITP process. As a result, we were able to obtain the same level of conduction as the cast bulk even at low temperature compression by the formation of direct contact between Al particles. This result can dramatically reduce the process temperature for bonding than process temperature (above 550 °C) for raw Al powder with the surface oxide layer. Such surface cleaning effect by the LO-ITP process is not limited to aluminum as long as the vapor pressure of the oxide is higher than the vapor pressure of the metal. In addition, different particle sizes of bare metal powder can be created by selecting the particle size of the raw material powder and controlling the process conditions of LO-ITP. Therefore, this process has the potential to achieve bonding that is not possible with traditional powder metallurgy processes, expanding the range of applications such as composite matrices, conductive ink or paste, and 3D printing.

## Figures and Tables

**Figure 1 materials-15-01553-f001:**
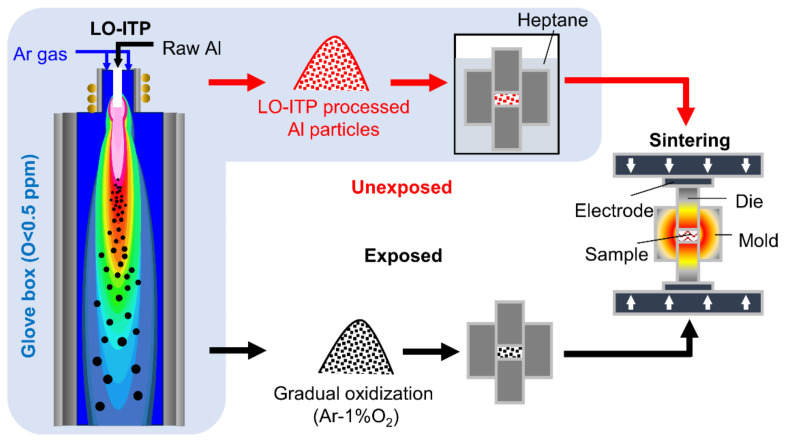
Schematic illustration of the experimental procedures. There were two routes. “Unexposed” indicates that the powder was not exposed to oxygen. “Exposed” indicates that the powder was gradually oxidized in a 1% O_2_ atmosphere.

**Figure 2 materials-15-01553-f002:**
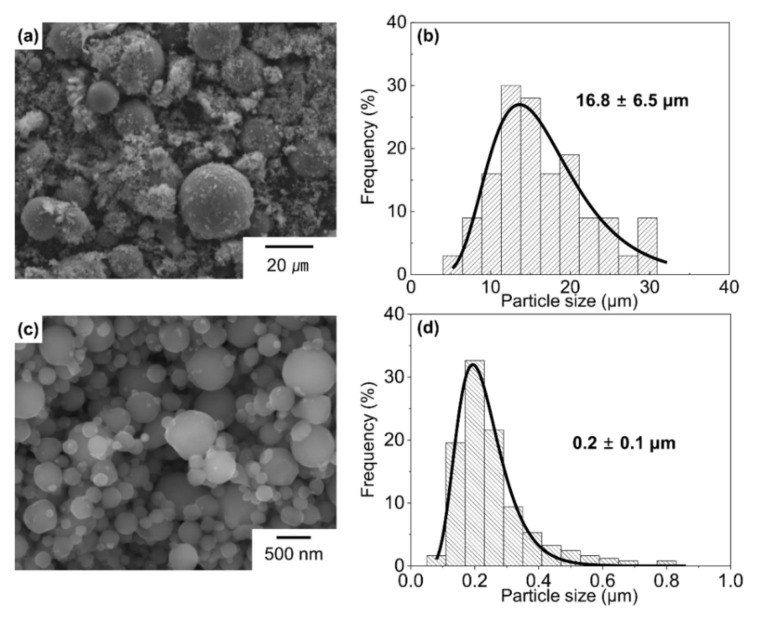
FE-SEM image of large Al particles fabricated from Al powder processed with LO-ITP at (**a**) low magnification. (**b**) Particle size distribution of large Al particles. FE-SEM image of fine Al particles fabricated from Al powder processed with LO-ITP at (**c**) high magnification. (**d**) Particle size distribution of fine Al particles (the average particle size and standard deviation were indicated in (**b**,**d**).

**Figure 3 materials-15-01553-f003:**
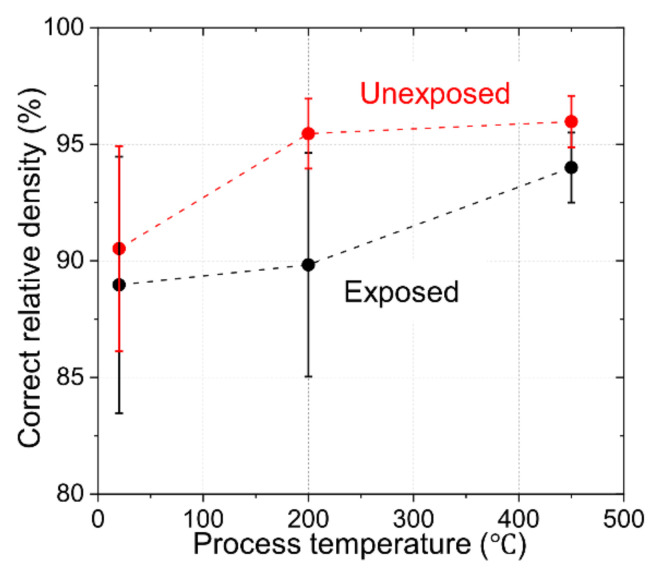
Correct relative density of unexposed and exposed Al compacts as a function of process temperature.

**Figure 4 materials-15-01553-f004:**
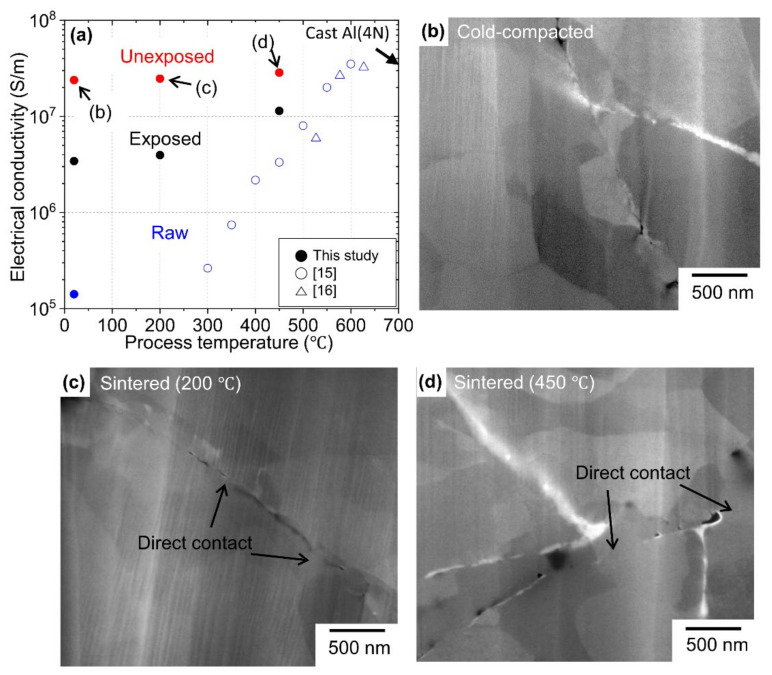
(**a**) Electrical conductivity as a function of process temperature for unexposed, exposed and raw Al compacts. FE-SEM images of unexposed compact Al that was (**b**) cold-compacted, (**c**) sintered at 200 °C and (**d**) sintered at 450 °C.

**Table 1 materials-15-01553-t001:** Density and electrical conductivity of unexposed and exposed Al compacts.

Sample	Process Temperature (°C)	Oxygen Content (wt%)	Absolute Density (g/cm^3^)	Correct Relative Density (%)	Electrical Conductivity (S/m)
Unexposed	20	1.4	2.46	90.5 ± 4.4	2.4∙10^7^
	200	1.1	2.53	95.5 ± 1.5	2.5∙10^7^
	450	0.7	2.50	96.0 ± 1.1	2.9∙10^7^
Exposed	20	2.5	2.39	89.0 ± 5.5	0.3∙10^7^
	200	2.2	2.52	89.8 ± 4.8	0.4∙10^7^
	450	1.3	2.64	94.0 ± 1.5	1.1∙10^7^

## Data Availability

Data sharing not applicable.

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
