# Peer review of "Surface Cleaning Effect of Bare Aluminum Micro-Sized Powder by Low Oxygen Induction Thermal Plasma"

_materials, 2022, doi:10.3390/ma15041553_

Round 1

Reviewer 1 Report

This research reports on the effect of surface cleaning on bare aluminum microsized powder using the process so-called low oxygen induction thermal plasma. To do so, the surface cleaning effect of Al particles was obtained by peeling the surface of Al particles using the LO-ITP method. The results demonstrated that direct contact was achieved at 450 °C sintering as per microstructure approach.

This is an interesting work which can potentially be published in Materials, but only after the following comments are all taken into consideration in the revised manuscript. At this stage, the comments and recommendations are as follows:

1) My major criticism is associated with the experimental apparatus and test procedures. The description of the test results leading to unexposed and exposed routes is ambiguous and unclear. A real test setup in which the procedures and instrumentations are identified is crucial in order to put the confidence in the readers that there ARE actually experimental investigation and the data obtained are reliable. 

2) Are the figure legend information (micrometers) in particle size vs frequency the indication of the data error?!  IT must be clarified. 

3) What is the advantage of thermal plasma over spark plasma? The authors must discuss this by comparing thermal plasma with spark plasma which is used in the following reference. It must be discussed in the Introduction portion of the manuscript.

"Process Optimization of In Situ Magnetic-Anisotropy Spark Plasma Sintering of M-Type-Based Barium Hexaferrite BaFe12O19", Materials, https://doi.org/10.3390/ma14102650 

4) Can the authors comment on why using commercially available Al particles necessitates removing the oxide layer on the surface? To my knowledge, it does not always lead to high conductivity. Please clarify.

5) Please discuss the novelty of your work in both the abstract and concluding remarks. 

In my point of view, if the above-mentioned comments are addressed carefully, the revised manuscript can then be reconsidered for publication in Materials. 

Author Response

Thank you so much for taking your precious time to review our manuscript and giving us positive comments and suggestions. We have revised the manuscript following your comments. The revised parts (figures or texts) were indicated using highlight in this review answer (Please see the attached file).

Reviewer 2 Report

The paper is corrected with same small corrections:

  • in same parts of paper (for l.158) we have "x". It should be "."
  • l.112 - the Authors write about Eq (1). But this Eq were not appers earlier. In this pleace it is necessery to put this Eq.
  • l.118 - pleace cut "Equation (2)"
  •  The names in reference [20] must be wrtitten as others name

Author Response

Thank you so much for taking your precious time to review our manuscript and giving us positive comments and suggestions. We have revised the manuscript following your comments. The revised parts (figures or texts) were indicated using highlight in this review answer (please see the attached file).

Reviewer 3 Report

The paper reports an interesting research about the improvement in the processability of Al powders. Research methodology is quite clear and the results are reported in a clear and easy to understand manner.

There are however some points which need to be better clarified before judging the paper ready for publication.

Line 104 and following (including Figure 1):

Is the process applied for the sintering of treated powders a SPS one? If Spark Plasma Sintering is applied, please provide a better description of the process parameters used.

Line 154 and following:

When evaluating the EC results of treated powders, a reference value is used quoting ref. [15 and 16]. Given the fact that ref [15] is rather dated, it is quite difficult to get a copy of it, so the reference to the granulometry cannot be verified. Are the granulometries of powders in [15], [16] and in the proposed investigation comparable? The term "commercial" powder may in fact be applied to several different products...

A similar observation may be applied to the lines 177-178 and to the conclusions (particularly lines 189-190)

Please clarify

Author Response

(The authors gave the same response as above.)

Round 2

Reviewer 1 Report

The authors have NOT responded to my comments satisfactorily. The following comments have been ignored:

1) My major criticism is associated with the experimental apparatus and test procedures. The description of the test results leading to unexposed and exposed routes is ambiguous and unclear. A real test setup in which the procedures and instrumentations are identified is crucial in order to put the confidence in the readers that there ARE actually experimental investigation and the data obtained are reliable. 

3) What is the advantage of thermal plasma over spark plasma? The authors must discuss this by comparing thermal plasma with spark plasma which is used in the following reference. It must be discussed in the Introduction portion of the manuscript.

"Process Optimization of In Situ Magnetic-Anisotropy Spark Plasma Sintering of M-Type-Based Barium Hexaferrite BaFe12O19", Materials, https://doi.org/10.3390/ma14102650 

4) Can the authors comment on why using commercially available Al particles necessitates removing the oxide layer on the surface? To my knowledge, it does not always lead to high conductivity. Please clarify.

I expect point by point response to my comments. 

Author Response

Thank you so much for taking your precious time to review our manuscript and giving us positive comments and suggestions. We have revised the manuscript following your comments. The revised parts (figures or texts) were indicated using highlight in this review answer (please see the attached file).

  1. My major criticism is associated with the experimental apparatus and test procedures. The description of the test results leading to unexposed and exposed routes is ambiguous and unclear. A real test setup in which the procedures and instrumentations are identified is crucial in order to put the confidence in the readers that there ARE actually experimental investigation and the data obtained are reliable.

Response)

The procedures of Al powder production (LO-ITP) and preparation for consolidation were conducted in glove box where the oxygen level is under 0.5 ppm in order to prevent oxidization of Al powder. Although we used a sintering machine outside from a glove box, the prepared mold set was transferred by a transfer vessel filled with heptane from the glove box in order to prevent further oxidization during transferring to the sintering machine. After finishing the sintering, the Al compact was gradually oxidized in the Ar-1%O2 atmosphere to avoid heating or fire by drastic oxidation on the surface or open pores inside the compacts. The Al compact prepared by such procedures (unexposed to the air) was called as “unexposed Al compact”.

On the other hand, the LO-ITP processed Al powder was gradually oxidized in Ar-1%O2 previously in order to expose it to the air from glove box, and then sintered. In this procedure, there was no usage of heptane for transferring the mold set containing the oxidized Al powder. The Al compact prepared by such procedures (exposed to the air) was called as “exposed Al compact”. The evaluation of electrical conductivity and/or microstructure observation were conducted under the same procedures and conditions.

Revised part)

  • Figure 1 was revised to give information about such experiment procedures in order to put the confidence in the readers.
  • Line 99-102: In the glove box where oxygen content was controlled under 0.5 ppm, the unexposed Al powder was poured into a tungsten cobalt mold, and the mold set was transferred to the current sintering machine using a transfer vessel filled with heptane to avoid further oxidization.

  1. Are the figure legend information (micrometers) in particle size vs frequency the indication of the data error?!  IT must be clarified.

Response)

In Figure 2, the average particle size obtained from particle size distribution was indicated. The values after ‘ ’ are standard deviation. In order to give information to the readers, the caption of Figure 2 was revised.

Revised part)

  • Line 142-146: Figure 2. FE-SEM image of large Al particles fabricated from Al powder processed with LO-ITP at (a) low magnification. (b) Particle size distribution of large Al particles. FE-SEM image of fine Al particles fabricated from Al powder –processed with LO-ITP at (c) high magnification. (d) Particle size distribution of fine Al particles (the average particle size and standard deviation were indicated in (b) and (d)).
  1. What is the advantage of thermal plasma over spark plasma? The authors must discuss this by comparing thermal plasma with spark plasma which is used in the following reference. It must be discussed in the Introduction portion of the manuscript. "Process Optimization of In Situ Magnetic-Anisotropy Spark Plasma Sintering of M-Type-Based Barium Hexaferrite BaFe12O19", Materials, https://doi.org/10.3390/ma14102650 

Response)

As you pointed out, the SPS process has the effect of breaking the oxide layer on the surface and promoting sintering. In this study, it is referred to as SPS, but it is actually “direct current sintering”. As a reference, the oxide layer could not be broken in the sample sintered at 400 °C using commercially available Al powder with an oxide layer. Therefore, in this experiment at 400 °C, the effect of removing the oxidized phase by SPS can be ignored, and only the surface cleaning effect of the thermal plasma process can be extracted.

  1. Can the authors comment on why using commercially available Al particles necessitates removing the oxide layer on the surface? To my knowledge, it does not always lead to high conductivity. Please clarify.

Response)

This study showed the electrical conductivity value of Al compact produced by LO-ITP processed Al powder in order to suggest the possibility of Al/Al direct contact at lower temperatures. The Al powder with less oxide layer on the surface can be applied to the matrix for composite materials, conductive ink, and 3D printing where low process temperature is desirable (please find line 198-200).

The oxide-free interfaces might not always lead to high conductivity if there are other obstacles of electrons such as dislocations or a large number of grain boundaries. However, the electrical conductivity necessarily deteriorates when the oxide layer resides at the interface between metal particles (please compare the electrical conductivity of unexposed, exposed and raw shown in Figure 4(a)). Especially, since Al oxide has 1019 times lower electrical conductivity compared to Al, the evaluation of electrical conductivity is the most convenient and efficient method to confirm the state of the oxide layer (for example, thickness, partial broken) in Al compact.

Revised part)

  • Line 157-158: Since the electrical conductivity of Al oxide is 10-19 times that of Al, the electrical conductivity was evaluated in order to confirm if the Al oxide layer is existed or not at Al/Al interfaces.

  1. Please discuss the novelty of your work in both the abstract and concluding remarks. 

Response)

The novelty of our work is that the surface cleaned metal powder can be produced by the LO-ITP technique, which is the original technique in our research group. In previous, easily oxidizable metal powder always has a contamination layer on the surface (oxide layer). The oxide layer prevents direct contact between metal inclusions. Therefore, easily oxidizable metal powder required a high bonding temperature. This is the reason why the application of the easily oxidizable metal powders has been limited in various industries where low bonding temperature is desirable such as composite materials, 3D printing, conductive ink. In other words, easily oxidizable metal powder also be widely applicable to such industries if surface oxide film could be removed.  

In previous, only partial direct contact or bonding of Al metal particles was reported after current sintering by broken oxide layer over 550 ℃. On the other hand, we demonstrated that the direct contact of Al particles could be formed at a lower sintering temperature (400 ℃) when using LO-ITP processed Al powder.

We revised the abstract and conclusions in order to deliver the novelty of our work well to the readers. Please reconsider the revised text.

Revised part)

  • Line 13-27: The development of bare metal powder is desirable for obtaining conductive interfaces by low-temperature sintering to be applied in various industries of 3D printing, conductive ink or paste. In our previous study, bulk Al made from Al nanopowder that was prepared with low oxygen thermal plasma (LO-ITP), which is original metal powder production technique, showed high electrical conductivity comparable to Al casting material. This study discusses the surface cleaning effect of Al particles expected to be obtained by peeling the surface of Al particles using the LO-ITP method. Bare metal microsized powders were prepared using LO-ITP by controlling the power supply rate and preferentially vaporizing the oxidized surface of the Al powder. Electrical conductivity was evaluated in order to confirm if there is the oxide layer at the Al/Al interface. The Al compact at room temperature made from LO-ITP processed Al powder showed an electrical conductivity of 2.9∙107 S/m, which is comparable to that of cast Al bulk. According to the microstructure observation, especially for the interfaces between bare Al powder, direct contact was achieved at 450 °C sintering. This process temperature is lower than the conventional sintering temperature (550 °C) of commercial Al powder without any surface cleaning. Therefore, surface cleaning using LO-ITP is the key to opening a new gate to the powder metallurgy process.

  • Line 194-207: We succeeded in obtaining the surface cleaning effect of Al microsized powder by preferentially evaporating the nonconductive surface oxide layer through the LO-ITP process. As a result, we were able to obtain the same level of conduction as the cast bulk even at low temperature compression by the formation of direct contact between Al particles. This result can dramatically reduce the process temperature for bonding than process temperature (above 550 °C) for raw Al powder with the surface oxide layer. Such surface cleaning effect by the LO-ITP process is not limited to aluminum as long as the vapor pressure of the oxide is higher than the vapor pressure of the metal. In addition, different particle sizes of bare metal powder can be created by selecting the particle size of the raw material powder and controlling the process conditions of LO-ITP. Therefore, this process has the potential to achieve bonding that is not possible with traditional powder metallurgy processes, expanding the range of applications such as composite matrices, conductive ink or paste, and 3D printing.

Reviewer 2 Report

I accept in present form

Author Response

Thank you very much for your kind comments.

Reviewer 3 Report

Authors clarified the different points. The paper is considered ready for publication

Author Response

(The authors gave the same response as above.)

Round 3

Reviewer 1 Report

In previous round of review, I asked authors to discus in the revised manuscript that what the advantage of thermal plasma over spark plasma is? I advised authors that they must discuss this by comparing thermal plasma with spark plasma which is used in the following reference in the Introduction portion of the manuscript. "Process Optimization of In Situ Magnetic-Anisotropy Spark Plasma Sintering of M-Type-Based Barium Hexaferrite BaFe12O19", Materials, https://doi.org/10.3390/ma14102650

The Authors responded the following: "As you pointed out, the SPS process has the effect of breaking the oxide layer on the surface and promoting sintering. In this study, it is referred to as SPS, but it is actually “direct current sintering”. As a reference, the oxide layer could not be broken in the sample sintered at 400 °C using commercially available Al powder with an oxide layer. Therefore, in this experiment at 400 °C, the effect of removing the oxidized phase by SPS can be ignored, and only the surface cleaning effect of the thermal plasma process can be extracted."

You have not included your response in the revised manuscript. You must compare your work with that work with appropriate referencing to that work and then include your response above there. 

Author Response

Thank you so much for taking your precious time to review our manuscript and giving us positive comments and suggestions. We have revised the manuscript following your comments. The revised parts (figures or texts) were indicated using highlight in this review answer.

Round 4

Reviewer 1 Report

The authors have satisfactorily responded to my comments.